# Semantic uncertainty intervals for disentangled latent spaces

Swami Sankaranarayanan[1], Anastasios N. Angelopoulos[2],
Stephen Bates[2], Yaniv Romano[3], and Phillip Isola[1]

[1]MIT
[2]University of California, Berkeley
[3]Technion—Israel Institute of Technology

## Abstract

Meaningful uncertainty quantification in computer vision requires reasoning about semantic information—say, the hair color of the person in a photo or the location of a car on the street. To this end, recent breakthroughs in generative modeling allow us to represent semantic information in disentangled latent spaces, but providing uncertainties on the semantic latent variables has remained challenging. In this work, we provide principled uncertainty intervals that are guaranteed to contain the true semantic factors for any underlying generative model. The method does the following: (1) it uses quantile regression to output a heuristic uncertainty interval for each element in the latent space (2) calibrates these uncertainties such that they contain the true value of the latent for a new, unseen input. The endpoints of these calibrated intervals can then be propagated through the generator to produce interpretable uncertainty visualizations for each semantic factor. This technique reliably communicates semantically meaningful, principled, and instance-adaptive uncertainty in inverse problems like image super-resolution and image completion. Code and demos can be found on our project page.

## 1 Introduction

When making decisions with visual data, such as automated vehicle navigation with blurry images, uncertainty quantification is critical. The relevant uncertainty pertains to a low-dimensional set of semantic properties, such as the locations of objects. However, there is a wide class of image-valued estimation problems—from super-resolution to inpainting—for which there does not currently exist a method of producing semantically meaningful uncertainties. As an example, imagine doing uncertainty quantification for medical image reconstruction from, say, a fast but undersampled MRI scan. In such a setting, pixelwise intervals [16, 40, 5] are not very useful. We need uncertainty on the underlying semantics—e.g., whether there is a tumor, and if so, of what size and shape.

We make progress on this problem by bringing techniques from quantile regression and distribution-free uncertainty quantification together with a disentangled latent space learned by a *generative adversarial network* (GAN). We call the coordinates of this latent space *semantic factors*, as each controls one meaningful aspect of the image, like age or hair color. We do not require these semantic factors to be statistically independent. Our method takes a corrupted image input and predicts each semantic factor along with an uncertainty interval that is guaranteed to contain the true semantic factor. When the model is unsure, the intervals are large, and vice-versa. By propagating these intervals through the GAN coordinate-wise, we can visualize uncertainty directly in image-space without resorting to per-pixel intervals—see Figure 1. The result of our procedure is a rich form of uncertainty quantification directly on the estimates of semantic properties of the image.

36th Conference on Neural Information Processing Systems (NeurIPS 2022).

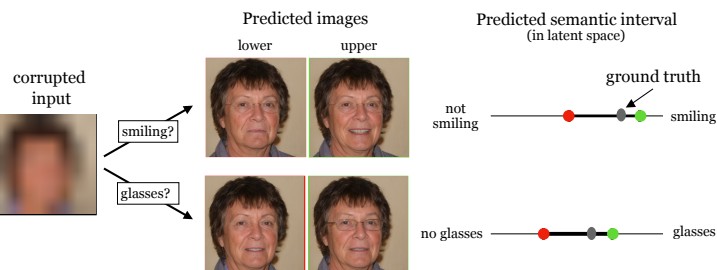

Figure 1: **Uncertainty intervals over semantic factors** produced by our method. We express the intervals in a disentangled latent space that allows us to factorize the uncertainty into meaningful components.

More concretely, we receive input images $X$ and then predict an *uncertainty interval* for each of $D$ *semantic factors* $Z_d$, $d = 1, ..., D$, which are the elements of a disentangled latent space. The method involves training an *encoder* (a neural network that takes images as input and produces outputs in the latent space of the GAN) on $(X, Z)$ pairs to give us three different outputs:

1. **The point prediction,** $f(X)$. This is the encoder's best guess at the semantic factors $Z$.

2. **The estimated lower conditional quantile,** $q_{\frac{\alpha}{2}}(X)$. The encoder believes that $q_{\frac{\alpha}{2}}(X)$ is a lower bound on the value of $Z$ given $X$.

3. **The estimated upper conditional quantile,** $q_{1-\frac{\alpha}{2}}(X)$. The encoder believes that $q_{1-\frac{\alpha}{2}}(X)$ is an upper bound on the value of $Z$ given $X$.

Once the above encoder is trained, as described in Section 2.2, we use it to form an uncertainty interval for each semantic factor. However, for the $d$th element of the latent code, the naive interval $(q_{\frac{\alpha}{2}}(X)_d, q_{1-\frac{\alpha}{2}}(X)_d)$ is not guaranteed to contain the ground truth value in finite samples. We propose to perform a calibration procedure to fix this problem, yielding the sets

$$\mathcal{T}(X)_d = \left[ q_{\frac{\alpha}{2}}^{\text{cal}}(X)_d, q_{1-\frac{\alpha}{2}}^{\text{cal}}(X)_d \right], \tag{1}$$

where $q^{\text{cal}}$ is a calibrated version of $q$ constructed using the tools in Section 2.3. Once we have done so, the intervals will contain $\alpha$ fraction of the true latent codes with high probability. In other words, for any user-chosen levels $\alpha$ and $\delta$, we can output intervals that with probability $1 - \delta$ satisfy

$$\mathbb{E}\left[ \frac{1}{D} \Big| \big\{ d : Z_d \in \mathcal{T}(X)_d \big\} \Big| \right] \geq 1 - \alpha, \tag{2}$$

for a new test point $(X, Z)$, regardless of the distribution of the data, the encoder used, and the number of data points used in the calibration procedure. This guarantee, described more carefully in Definition 2.1, says that the intervals cover $1 - \alpha$ fraction of the semantic factors unless our calibration data is not representative of our test data (which only happens with a probability $\delta$ which goes to 0 as the number of calibration data points grows). We visualize each of the $d \in \{1, ..., D\}$ intervals in latent space by propagating the $d$th lower and upper endpoints through the generator with all other entries in the latent fixed to the point prediction (see Section 2.3 for a formal explanation).

## 1.1 Central Contribution

To our knowledge, this is the first algorithm for uncertainty intervals on a learned semantic representation with formal statistical guarantees. By propagating these intervals through the generator, we are able to visualize uncertainty in an intuitive new way that directly encodes semantic meaning. This is an important step towards interpretable uncertainties in general image-valued estimation problems. However, the reader should note that our technique requires access to a disentangled latent space, such as that of a StyleGAN, and provides no guarantees about the degree of disentanglement.

## 2 Method

### 2.1 Notation and goal

Our data consist of pairs $(X, Z)$—the corrupted image $X$ in $\mathcal{X} = [0, 1]^{H \times W}$, and the latent code $Z \in \mathcal{Z}$, where $\mathcal{Z} = \mathbb{R}^D$. As mentioned in the introduction, we think of $Z$ as a disentangled representation with $d$ *semantic factors*—*i.e.*factors of variation corresponding to interpretable features in an image, such as hair color and expression. For simplicity, assume each of the $d$ dimensions controls a single semantic factor; in practice, we ignore those that do not.

In our sampling model, $X$ is generated from $Z$ by composing two functions. The first function is a fixed generator $G : \mathcal{Z} \to \mathcal{Y}$, where $\mathcal{Y} = [0, 1]^{H \times W}$, which takes the latent vector $Z$ and produces the ground truth image $Y \in \mathcal{Y}$ (for ease of notation, we assume $X$ and $Y$ have the same shape). The second function is a corruption model, $F : \mathcal{Y} \to \mathcal{X}$, which degrades the ground truth image $Y$ to produce the corrupted image $X$, for example by randomly masking out part of the image. To summarize our data-generating process, we have

$$Y = G(Z) \text{ and } X = F(Y); \tag{3}$$

**Goal #1.** Our first goal is to train an encoder $E$ to recover $Z$ from $X$—in other words, to invert the mapping $F \circ G$—with a heuristic notion of uncertainty. The encoder's point prediction will be a function $f : \mathcal{X} \to \mathcal{Z}$. The uncertainty will be parameterized by two functions, $q_{1-\frac{\alpha}{2}} : \mathcal{X} \to \mathcal{Z}$ and $q_{\frac{\alpha}{2}} : \mathcal{X} \to \mathcal{Z}$, denoting our estimates of the $1 - \frac{\alpha}{2}$ and $\frac{\alpha}{2}$ conditional quantiles, respectively. These conditional quantiles are potentially bad estimates; they do not natively possess the statistical guarantee we desire.

**Goal #2.** Having trained the encoder and the conditional quantile estimates, we will output uncertainty intervals in the disentangled latent space. Each dimension will get its own interval, which has the form in (2). Ultimately, our uncertainty intervals will come with a statistical risk control guarantee, like in (2), that is distribution-free—i.e., valid irrespective of the model or data distribution. The risk function controlled is the false negative rate *in the latent space*. In other words, we bound the fraction of semantic factors not covered by their calibrated intervals above by $\alpha$ (say, $\alpha = 10\%$).

**Definition 2.1** (Risk-Controlling Prediction Set (RCPS)). A set-valued function $\mathcal{T} : \mathcal{X} \to 2^{\mathcal{Z}}$ is an $(\alpha, \delta)$ risk-controlling prediction set if

$$\mathbb{P}\Big(\mathbb{E}\big[L(\mathcal{T}(X), Z)\big] > \alpha\Big) \leq \delta, \tag{4}$$

where

$$L(\mathcal{T}(X), Y) = 1 - \frac{\left|\big\{d : Z_d \in \mathcal{T}(X)_d\big\}\right|}{HW}. \tag{5}$$

The reader should note here that the function $\mathcal{T}$ depends on the calibration data. The outer probability in (4) is over the randomness in this calibration procedure; the inner expectation is over the new test point, $(X, Y)$. The reader should note that in our setting, because we can generate infinite data from the GAN, it is always possible to take $\delta$ arbitrarily close to 0, effectively making $\mathcal{T}$ nonrandom. In the two following subsections, we address each of our goals separately.

### 2.2 Goal #1: Training the encoder for quantile regression

Our job in this subsection is to learn the three functions $f$, $q_{\frac{\alpha}{2}}$, and $q_{1-\frac{\alpha}{2}}$. We do so by training a neural network with three different loss functions, one for each of three linear heads on top of the same feature extractor—see Figure 2 for the training protocol.

**Loss function for point prediction.** We supervise the point prediction with three loss functions. The first is an $L_1$ loss directly in the latent space, and encourages $f(X)$ to be close to $Z$.

$$\mathcal{L}_1\big(f(x), z\big) = \big|\big|f(x) - z\big|\big|_1 \tag{6}$$

The second loss term imposes a *domain specific prior* on the generated image $G(f(X))$. For our experiments on faces, we use an identity loss which encourages $G(f(X))$ to contain attributes that have the same "identity" — or semantic content — as $Y$. We calculate the identity loss using a

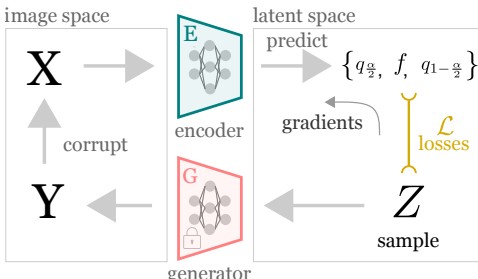

Figure 2: **Our training pipeline**, visualized above, shows our training process for the prediction $\hat{f}$, lower quantile $q_{\frac{\alpha}{2}}$ and the upper quantile $q_{1-\frac{\alpha}{2}}$.

pretrained embedding function, $\text{ID} : \mathcal{Y} \to \mathbb{R}^{d'}$ for some $d'$, which projects $Y$ to an embedding space where images with different identities land far away from one another. Finally, we calculate our loss using cosine similarity,

$$\mathcal{L}_{\text{ID}}(x, y) = \frac{\langle \text{ID}(x), \text{ID}(y) \rangle}{||\text{ID}(x)|| \cdot ||\text{ID}(y)||}, \tag{7}$$

The third is a *perceptual loss* on the generated image $G(f(X))$ that imposes similarity between $G(f(X))$ and $X$ in the embedding space of a large pretrained image classification model. We impose perceptual similarity by computing the LPIPS loss [50] which has been shown to preserve image quality [17]. The perceptual loss is calculated using a pretrained feature extractor in a similar manner to Eq 7. Following standard practice, the feature extractor used in this case is a VGG network, pretrained on Imagenet.

Finally, we combine the loss functions to form the loss function for the prediction $f$,

$$\mathcal{L}_{\text{pred}}(f(x), z) = \mathcal{L}_1(f(x), z) + c_1 \mathcal{L}_{\text{ID}}(G(f(x)), G(z)) + c_2 \mathcal{L}_{\text{LPIPS}}(G(f(x)), G(z)), \tag{8}$$

where $c_1$ and $c_2$ are hyper-parameters whose values are chosen based on a held out set.

**Loss function for quantile regression.** Quantile regression [30, 13, 32, 27, 28, 29, 31] is a statistical method for estimating the conditional quantiles of a distribution. The key idea of quantile regression is to supervise the regressor using a *quantile loss*,

$$\mathcal{L}_{\text{q}}^{\beta}(q_{\beta}(x), z) = (z - q_{\beta}(x))\beta \mathbb{1}\{z > q_{\beta}(x)\} + (q_{\beta}(x) - z)(1 - \beta)\mathbb{1}\{z \le q_{\beta}(x)\}. \tag{9}$$

The minimizer of the quantile risk is the true $\beta$ conditional quantile of $Z|X$. We supervise our conditional quantile estimates $q_{\frac{\alpha}{2}}$ and $q_{1-\frac{\alpha}{2}}$ with two separate instances of the quantile loss, $\mathcal{L}_{\text{q}}^{\frac{\alpha}{2}}$ and $\mathcal{L}_{\text{q}}^{1-\frac{\alpha}{2}}$ respectively.

This concludes our explanation of the model training procedure, summarized in Algorithm 1. Experimental details, such as the particular model architecture we use, are available in Section 3.

---

**Algorithm 1** Quantile GAN encoder training

---

   **Input:** training dataset $\mathcal{D}$, risk level $\alpha$, number of epochs $E$, fixed generator $G$
   **Output:** trained functions $f, q_{\frac{\alpha}{2}}$, and $q_{1-\frac{\alpha}{2}}$.
   $f, q_{\frac{\alpha}{2}}$, and $q_{1-\frac{\alpha}{2}} \leftarrow$ random model initialization
   **for** $e = 1$ **to** $E$ **do**
      `loss` $\leftarrow 0$
      **for** $(X^{\text{train}}, Z^{\text{train}})$ **in** $\mathcal{D}$ **do**
         `L1` $\leftarrow \mathcal{L}_{\text{pred}}(f(X^{\text{train}}), Z^{\text{train}})$
         `L2` $\leftarrow \mathcal{L}_{\text{q}}^{\frac{\alpha}{2}}(q_{\frac{\alpha}{2}}(X^{\text{train}}), Z^{\text{train}})$
         `L3` $\leftarrow \mathcal{L}_{\text{q}}^{1-\frac{\alpha}{2}}(q_{1-\frac{\alpha}{2}}(X^{\text{train}}), Z^{\text{train}})$
         `loss += L1 + L2 + L3`
      **end for**
      `loss.backward()`
   **end for**

---

## 2.3 Goal #2: Calibration

Having trained the model, we now calibrate it to achieve the statistical guarantee in Definition 2.1 using a set of calibration data $\{(X_i, Z_i)\}_{i=1}^{n}$ generated from the model and the upper-confidence bound procedure from [8]. The output of the procedure will be the function $\mathcal{T}$ from (2); specifically, we will learn the calibrated conditional quantiles $q_{\frac{\alpha}{2}}^{\text{cal}}$ and $q_{1-\frac{\alpha}{2}}^{\text{cal}}$.

Our procedure will calibrate the conditional quantiles by rescaling their size multiplicatively. We will ultimately choose a multiplicative factor $\hat{\lambda}$ that gives us the desired guarantee. Towards that end, we index a family of uncertainty intervals scaled by a free parameter $\lambda$ for each semantic factor,

$$\mathcal{T}_\lambda(X)_d = \left[ f(X)_d - \lambda\big(f(X)_d - q_{\frac{\alpha}{2}}(X)_d\big)_+, \quad f(X)_d + \lambda\big(q_{1-\frac{\alpha}{2}}(X)_d - f(X)_d\big)_+ \right]. \quad (10)$$

When $\lambda$ grows, the interval $\mathcal{T}_\lambda(X)_d$ also grows, and thus, the loss function $L(\mathcal{T}_\lambda(X), Y)$ shrinks. Therefore, by taking $\lambda$ large enough, we can always ensure the loss is zero. The challenge ahead is to pick $\hat{\lambda}$ to be the smallest value such that $\mathcal{T}_{\hat{\lambda}}$ is an RCPS as in (4).

The algorithm for selecting $\hat{\lambda}$ involves forming an upper confidence bound (UCB) for the risk, then picking the smallest value of $\lambda$ such that the upper confidence bound falls below $\alpha$. We give Hoeffding's UCB below, although we use the stronger Hoeffding-Bentkus bound from [8] in practice:

$$\hat{R}^+(\lambda) = \frac{1}{n} \sum_{i=1}^{n} L\left(\mathcal{T}_\lambda(X_i), Y_i\right) + \sqrt{\frac{1}{2n} \log \frac{1}{\delta}}. \quad (11)$$

Note that in our setting, we can always generate enough samples to drive the last term to $0$ for any $\delta$; however, if we only had a finite sample from some population, this would not be the case. We can then select $\hat{\lambda}$ by scanning from large to small values, $\hat{\lambda} = \min\left\{ \lambda : \hat{R}^+(\lambda') \leq \alpha, \ \forall \alpha' \geq \alpha \right\}$, or running binary search.

**Proposition 2.2** ($\mathcal{T}_{\hat{\lambda}}$ is an RCPS [8]). *With $\hat{\lambda}$ selected as above, $\mathcal{T}_{\hat{\lambda}}$ satisfies Definition 2.1.*

For the proof of this fact, along with a discussion the tighter confidence bounds used in our experiments and extensions to the underlying theory, see [8] and [3].

Having proven that $\mathcal{T}_{\hat{\lambda}}$ is an RCPS, we can simply set $\mathcal{T}(X) = \mathcal{T}_{\hat{\lambda}}(X)$ in (2); in other words, we set $q_{\frac{\alpha}{2}}^{\text{cal}}(X)_d = f(X)_d - \hat{\lambda}\big(f(X)_d - q_{\frac{\alpha}{2}}(X)_d\big)_+$ and $q_{1-\frac{\alpha}{2}}^{\text{cal}}(X)_d = f(X)_d + \hat{\lambda}\big(q_{1-\frac{\alpha}{2}}(X)_d - f(X)_d\big)_+$.

**Visualizing uncertainty intervals in image space**

We briefly describe our method for visualizing latent-space uncertainty intervals. In order to see the effect of a single semantic factor, we set it to either the lower or upper quantile and hold the other factors fixed to the point prediction. More specifically, for a particular dimension $d \in \{1, ..., D\}$, define the following vector:

$$\hat{Z}_k^d = \big(f(X)_1, ..., f(X)_{d-1}, \quad q_k^{cal}(X)_d, \quad f(X)_{d+1}, ..., f(X)_D\big). \quad (12)$$

We visualize the lower and upper quantiles in image space as $G(\hat{Z}_{\frac{\alpha}{2}}^d)$ and $G(\hat{Z}_{1-\frac{\alpha}{2}}^d)$ respectively. Since each semantic factor corresponds to an attribute, visualizing the lower and upper quantiles per-factor gives interpretable meaning to the latent-space intervals. For example, in Figure 1, the images of the child smiling give a range of possible expressions the model believes are consistent with the underlying image.

## 3 Experiments

### 3.1 Dataset descriptions

**FFHQ** We use the StyleGAN framework pretrained using the Flickr-Faces-HQ (FFHQ) dataset [25]. FFHQ is a publicly available dataset consisting of 70,000 high-quality images at $1024 \times 1024$ resolution. The data used to train the quantile encoder and for the experiments in this section is sampled from the generator pretrained on FFHQ.

**CelebA-HQ** We use the CelebA-HQ dataset [23] to demonstrate the effectiveness of our approach on real world data. The dataset contains 30,000 high-quality images of celebrity faces. Since the real dataset is used only for evaluation, we use images from standard test split.

**CLEVR.** In order to have a controlled setup where we can easily identify disentangled factors of variation, we generate synthetic images of objects based on the CLEVR dataset [22]. This dataset provides a programmatic way of generating synthetic data by explicitly varying specific semantic factors. We create a synthetic dataset by varying $\{color, shape\}$ and fixing the other factors such as lighting, material and camera jitter.

### 3.2 Experimental setup

**Model architectures** In all our experiments, we use the StyleGAN2 [24] framework for the generator architecture $G$. For the experiments involving faces, we use the pretrained model available from the official repository. For the CLEVR-2D experiments, we train a simpler variant of StyleGAN2 generator from scratch. For the quantile regression, we use a standard architecture: the encoder network consists of a ResNet-50 backbone [19] with the final layer branching into the point prediction and conditional quantiles. The branching module is a standard combination of convolution and activation blocks followed by a fully connected layer of the expected output dimension; we call these the *heads* of the model. Specific details of the model architecture are provided in the supplementary material.

**Model training** We start by pretraining the generative model or acquiring an off-the-shelf pretrained generative model for the task at hand. In generative models such as StyleGAN, the style space that offers fine grained control over image attributes, is very high dimensional. From this high dimensional space, we extract only the disentangled dimensions following previous work on style space analysis [42]. In order to better focus the encoder's capacity only on the disentangled dimensions, we mask out the irrelevant dimensions for applying the quantile loss. However, the pointwise loss in (6) is applied to the full style vector to ensure that the pointwise prediction is able to match the true latents accurately, while the quantile heads focus on learning variablity only in the disentangled dimensions.

During the encoder training 2.2, the generative model $G$ is held frozen and only the parameters of the encoder $E$ are updated. The point prediction and conditional quantile heads are trained jointly with the Ranger optimizer [42] and a flat learning rate of 0.001 for all our experiments. The hyperparameter weights (Eq 8) are set to $c_1 = c_2 = 10.0$.

For the image super-resolution training, we augment the input dataset by using different levels of downsampled inputs, *i.e.*, we take the raw input and apply a random downsampling factor from $\{1, 4, 8, 16, 32\}$ and resize it to the original dimensions. For the image inpainting task, we vary the difficulty by choosing a random threshold to create the mask – lower thresholds implies fewer pixels are masked and vice-versa. The mask is concatenated to the image resulting in a $C + 1$-channel input to the encoder, where $C$ is the number of image channels. The detailed description of the mask generation procedure can be found in the supplementary material.

**Calibration and Evaluation** For both the synthetic object experiments with CLEVR and face experiments with FFHQ, we train the quantile encoder on data points sampled from the latent space of the pretrained generative model. This ensures that we have access to the *true* latents that resulted in each image. We generate 100k samples per model and generate a random 80-10-10 split for training, calibration and validation.

### 3.3 Findings

In the following experiments, we explore different properties of our intervals. The problem types include image super-resolution and image inpainting. The risk level $\alpha$ and the user-specified error threshold $\delta$ are fixed to 0.1, unless specified otherwise.

#### 3.3.1 Producing semantic uncertainties

**Goal.** We qualitatively verify that the proposed approach outlined in Section 2 produces visually meaningful uncertainties.

**Description.** We train a quantile encoder with Algorithm 1. Then we generate $n = 5000$ images by sampling latents and propagating them through the encoder-generator combination. We use these $n$ images as calibration data for the procedure in 2.3. Finally, we randomly sample a new test point, pass it through the calibrated encoder, and form the uncertainty intervals in image space as in Section 2.3.

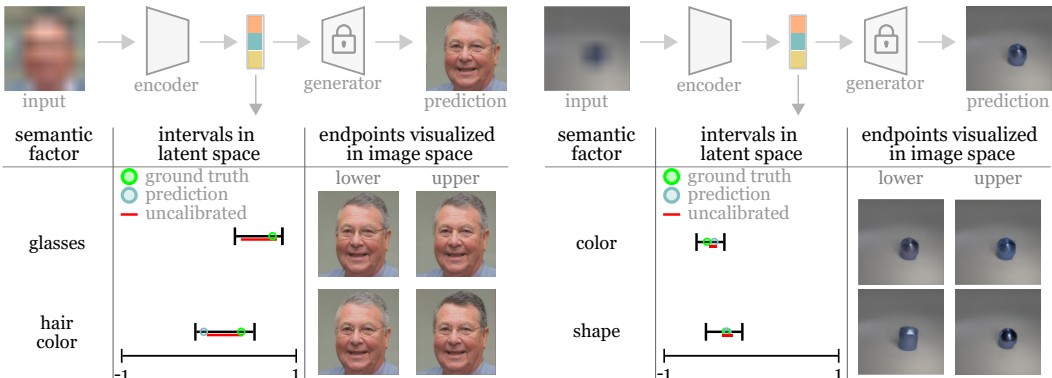

Figure 3: **Semantically meaningful uncertainty intervals** produced by our method on example images sampled from the generator trained on the FFHQ dataset (left) and CLEVR dataset (right). The corrupt image is provided as input to the encoder which outputs a pointwise prediction and quantile predictions for each style dimension. We plot the calibrated and uncalibrated intervals as well as their visualizations in image-space. [Best viewed in color, zoom in for detail.]

**Results.** The results are illustrated in Figure 3 on images sampled from the generator trained on FFHQ (left) and CLEVR (right). In case of the FFHQ image, the person is wearing glasses in the lower quantile image and not in the upper; hence, the model is not certain that the person is wearing glasses. The model also expresses some uncertainty about the amount of gray versus brown hair. This outcome was predictable from the input image, where the fact that the person is wearing glasses is not obvious, and there is some hair color ambiguity. The results on the CLEVR dataset are analogous. The lower and upper quantile images yield similar colors, which is predictable from the blurry input. The model predicts that both a cylinder and sphere would be consistent with this blurry input. The calibrated quantiles cover the ground truth color value, while the uncalibrated ones do not.

### 3.3.2 Experiments with real data

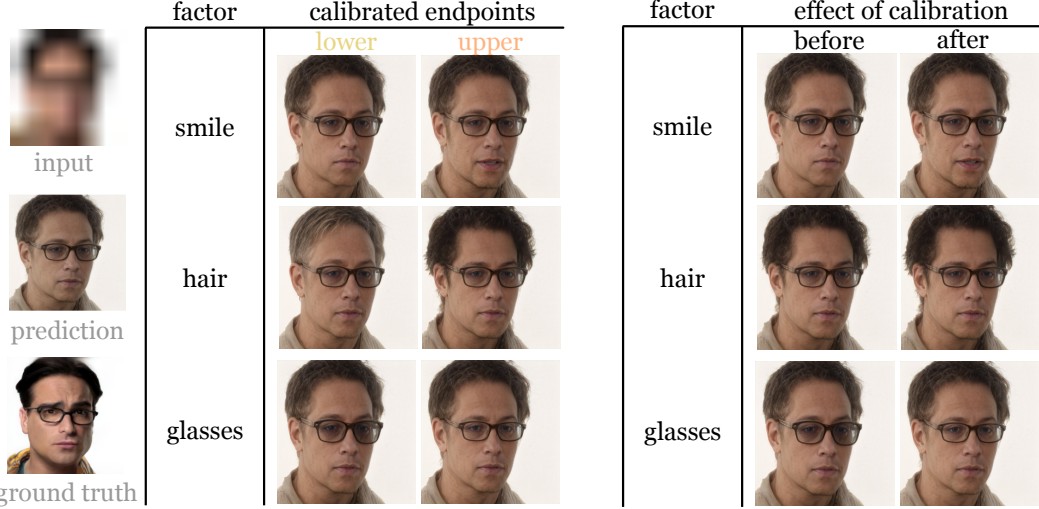

Figure 4: **Uncertainty intervals for real data** We demonstrate the outputs of our approach for a real image sampled from the CelebA-HQ faces dataset. **[Left]** The input is corrupted by downsampling the true image by a factor of 32. Though the prediction does not match the ground truth, the uncertainty captured by the quantiles remains meaningful. **[Right]** For the same input, we show the effect of our calibration procedure by visualizing the edges of the quantile intervals before and after calibration. Observe that in some cases, such as *smile* and *glasses*, there are clear visible changes due to the calibration approach. In other cases such as *hair*, the changes are not visually explicit. More such examples can be found in the supplementary material. [Best viewed in color. Zoom in for detail]

**Goal.** We calibrate our encoder with a combination of fake data sampled from the pretrained StyleGAN model and real data sampled from the CelebA-HQ face dataset for the image resolution task. The objective is to test to what extent we can extract meaningful uncertainty intervals for real data without explicitly training with them.

**Description** The difficulty with calibrating using real data is that the targets i.e the true style vectors that generated the real images are not available to us. To circumvent that difficulty, we back-project the real images into the latent space of the GAN using existing approaches to GAN inversion [24, 1]. Even though the mapping from the GAN latent space to the output is not one-one, the back-projection optimization procedure is able to visually match the appearance of the real images convincingly. This procedure provides us with the estimated "true" style vectors for each image in the real calibration set. We combine this with the generated data and perform the RCPS calibration procedure as described in Section 2.3. Note that, our encoder was only trained on generated data i.e. the same model as used in Section 3.3.1, we use the real data only for the calibration procedure.

**Qualitative results** The predicted quantiles and the effect of the calibration procedure for a real test image drawn from the CelebA-HQ dataset is visualized in Figure 4. The pointwise prediction of the encoder does not match the identity of the true image due to the high level of input corruption. However, using our approach, we are able to visualize the predicted uncertainty intervals in a meaningful manner. The visual effect of our calibration procedure is demonstrated on the right. Notice that some attributes such as *smile* and *glasses* have a visible change but others do not. The change is due to the fact that the calibration procedure adjusts the intervals in such a way to satisfy the required coverage guarantee.

### 3.3.3 Exploratory results with purposeful corruptions

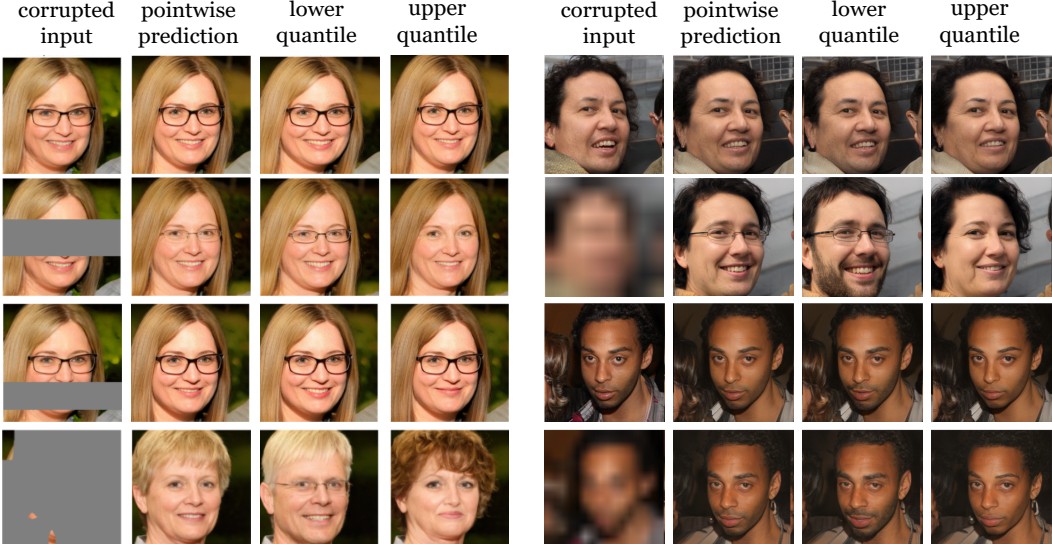

Figure 5: **Visualizing adaptivity [Left]** A random mask is applied to the same input image in each row. When there is no mask (1st row), the lower and upper quantiles are extremely close to the pointwise prediction. As we increase the regions that are being masked, the intervals predicted by the quantile encoder expand, as indicated by the variability on the lower and upper quantile predictions. **[Right]** We show the results of the encoder on two sets of images. The corruption intensity is varied across each set, the input image in the top row is not corrupted while the input in the bottom row is under-sampled by 16x. In both case, we can observe that the most diverse prediction is in the bottom row where the input is corrupted the most. [Best viewed in color. Zoom in for detail]

**Goal.** We probe our procedure to see if it will have the expected qualitative behavior.

**Description of experiment.** We sampled images from the held out set of the FFHQ pretrained GAN and applied purposeful corruptions to check if the resulting quantile estimates had semantic meaning. Both image resolution and image masking were used as corruption models. We qualitatively analyze

the results by visualizing the predictions. We include a quantitative measurement of variability estimated using image based metrics in the supplementary material.

**Qualitative results.** Figure 5 shows the results of this experiment for Image inpainting (left) and super-resolution (right). In the inpainting case, when nothing is masked, the quantiles are roughly identical. When the eyes are masked, the quantiles indicate the model does not know if the person was wearing glasses. When the mouth is masked, the model expresses uncertainty as to whether the woman is showing her teeth in the smile. Finally, when almost everything is masked, the quantile images are very different, representing individuals with entirely different identities. Similar behavior can be observed in the super-resolution case. The results are shown for two separate inputs; in both cases, the input in the top row is uncorrupted while in the bottom row, it is undersampled by 16x. The model is able to predict almost perfectly in the absence of corruption. The quantile predictions are extremely close as well. In the presence of corruption, both the pointwise prediction is off (as expected) and the quantile edges display much higher variability including in attributes such as hair shape, glasses, facial hair and perceived gender. The results from both these experiments point to an expected qualitative behavior of our proposed approach: the model exhibits more uncertainty with increased information loss at the input.

## 4 Interval sizes as a function of problem difficulty

**Goal.** We seek to construct intervals that adapt to the uncertainty of the input, *i.e.*, result in lower values for easier inputs and higher values for harder inputs. In this experiment we verify quantitatively how informative our intervals are as a function of increasing input corruption.

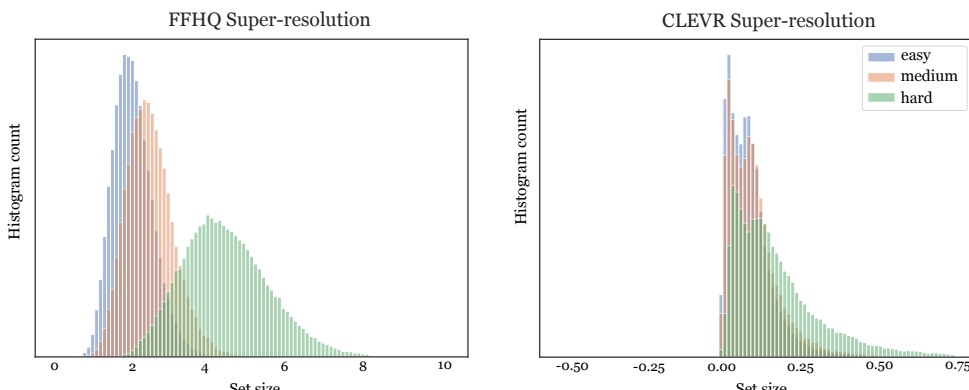

Figure 6: **Adapting to varying corruption levels:** Distribution of set-sizes for different input corruption levels for super-resolution on FFHQ and CLEVR.

**Description of experiment.** To simulate varying levels of corruption for image super-resolution, we create the difficulty levels {easy, medium, hard} that correspond to {1x, 8x, 32x} downsampled versions. All downsampled versions are resized to the same dimensions before presenting to the encoder. The results are computed on a set of 2000 images sampled from the validation split of the CelebA-HQ dataset. In order to obtain the set size for each input, we scale the quantile width using the threshold obtained after the RCPS calibration procedure.

**Results.** Figure 6 shows the set sizes for the super-resolution corruption model as a function of problem difficulty on two datasets, FFHQ and CLEVR. As expected, the set sizes increase with increasing problem difficulty indicating increasing uncertainty as corruption level increases.

## 5 Related Work

**GANs for Inverse problems.** The remarkable image generation properties of recent approaches such as BigGAN [11] and StyleGAN [24] has led to the increasing use of these models to solve inverse problems relating to image restoration such as image super-resolution and completion. All prior methods that use GANs for inverse problems such as the ones that use the pretrained generative model as an image prior [46, 9, 38, 36] or the ones that train an encoder model to project the input

into the generator's latent space [51, 42, 45, 41] focus on the accuracy of the point estimate but not on the uncertainty level of the input.

**GANs for Interpretability.** Despite providing no guarantees of image likelihood, unlike others such as Normalized Flow [26] or Score-based models [52], GANs have been used to develop interpretable approaches to image generation. The widespread use of GANs as opposed to other generative models in the interpretability is done due to the availability of an disentangled latent space [18, 49], which is a property we utilize in our work.

**Quantile Regression.** Quantile regression was first proposed in [30]. Since then, many papers have used the technique, applying it to machine learning [21, 37, 39], medical research [7], and more. Most relevant to us is conformalized quantile regression [43], which gives quantile regression a marginal coverage guarantee using conformal prediction. Our work instead uses risk-controlling prediction sets, a different distribution-free uncertainty quantification technique.

**Conformal prediction and distribution-free uncertainty quantification.** At the core of our proposed method is the distribution-free, marginal risk-control technique studied in [8] and [3]. These ideas have their roots in the distribution-free marginal guarantees of conformal prediction, proposed in [48]. Conformal prediction is a flexible method for forming prediction sets that satisfy a marginal coverage guarantee, under no assumptions besides the exchangeability of the test point with the calibration data [48, 47, 34, 35, 33, 2]. Conformal prediction has been studied in computer vision [20, 12, 6, 44, 4], natural language [14], drug discovery [15], criminal justice [10], and more.

We are not aware of work applying the notions of conformal prediction and quantile regression to the latent space in generative models.

## 6 Conclusion

Experiments indicate that with an appropriately disentangled model, latent space uncertainty intervals express a useful semantic notion of uncertainty previously unavailable in computer vision. Limitations of our method include that the calibration data must be reflective of the data distribution, and that we assume access to a disentangled latent space. We see our work as part of a larger tapestry of results in generative models, and our technique will remain applicable as progress gets made on disentanglement and backprojection. Furthermore, by introducing a notion of statistical rigor to GAN based approaches, our work creates a path for application of such models to safety critical settings.

## 7 Ethics

The ethics of generative modeling itself has been called into question given recent events, *e.g.*, the development of deep fakes. Nonetheless, we believe the downstream consequences of this work will likely be positive. The techniques herein do not change the predictions of a generative model; they simply provide a calibrated notion of its uncertainty in a relevant semantic space. Thus, the standard criticism of generative modeling—that it will enable widespread deep fakes—is not applicable. Furthermore, we expect having a statistically valid and semantically rich notion of uncertainty will provide a sobering reliability assessment of these models, perhaps mitigating the chance of harmful failures. Finally, although we use face datasets due to their ubiquity in this literature, we have attempted to ethically treat topics like gender and race where they arise.

**Acknowledgements**

A. N. A. was supported by NSF GRFP. S. B. was supported by the Foundations of Data Science Institute and the Simons Institute. Y. R. was supported by the Israel Science Foundation (grant No. 729/21). Y. R. thanks the Career Advancement Fellowship, Technion, for providing research support. S. S. and P. I. 's research for this project was sponsored by the United States Air Force Research Laboratory and the United States Air Force Artificial Intelligence Accelerator and was accomplished under Cooperative Agreement Number FA8750-19-2- 1000. The views and conclusions contained in this document are those of the authors and should not be interpreted as representing the official policies, either expressed or implied, of the United States Air Force or the U.S. Government. The U.S. Government is authorized to reproduce and distribute reprints for Government purposes notwithstanding any copyright notation herein. S. S. also acknowledges the MIT SuperCloud and Lincoln Laboratory Supercomputing Center for providing compute resources that have contributed to the results reported in this work.

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
