# A    Model architectures

## A.1    Face experiments

For the encoder, we use a resnet-50 backbone followed by projection heads that output pointwise, lower and upper quantile predictions. Each projection head consists of a convolution layer followed by a Leaky-Relu activation and a global average pooling layer. The input to each projection head is the output of the backbone network – a feature map of size $512 \times 4 \times 4$ and the output dimension is the number of style dimensions – in the case of the pretrained FFHQ styleGAN2 used in our experiments, this value is 9088.

For the generator, we use a FFHQ pretrained styleGAN2 trained to output faces of resolution $1024 \times 1024$ obtained from the official implementation. No discriminator is used during training.

## A.2    CLEVR experiments

For the encoder, we use a resnet-18 backbone followed by projection heads that output pointwise, lower and upper quantile predictions. Each projection head consists of a convolution layer followed by a Leaky-Relu activation and a global average pooling layer. The input to each projection head is the output of the backbone network – a feature vector of size $512$ and the output dimension is the number of style dimensions – in the case of the pretrained CLEVR styleGAN2 used in our experiments, this value is 204.

For the generator, we use a modified version of styleGAN2 that is trained to output images of resolution $128 \times 128$. In order to have a controlled latent space, we reduce the size of the style vectors from $512$ in the original model to $12$. This was done to reduce the size of the resulting style dimension from 9088 to 204. Since the model was trained on the CLEVR dataset which has less variability compared to other datasets such as FFHQ, the model was able to converge successfully even at this reduced capacity.

# B    Training details

## B.1    Input preprocessing

For the face experiments, the inputs to the encoder are resized to $256 \times 256$ and are rescaled to $[-1, 1]$ range. For the super-resolution experiment, the original input is first downsampled as required (i.e. 8x/16x etc) and is resized back to the input resolution $256 \times 256$. For the image inpainting experiment, the corruption mask is generated using the procedure outlined in Section B.2. The image input is then masked to only expose the unmasked parts – hence the corruption; the mask is concatenated along with the image as an additional input. Example of a masked image is shown the main manuscript in Figure 4.

The procedure described above is repeated for the CLEVR epxeriments with the exception that the input size is $128 \times 128$.

## B.2    Mask generation procedure for image inpainting

For generating a corruption model for image completion, we generate a binary mask in a controlled manner. For each input image of size $H \times W \times C$, we start by generating a random mask of size $H \times W$ where each pixel value in contained in the interval $[0, 1]$. For each difficulty level as mentioned in the manuscript (*easy, medium, hard*), we activate only those pixels in the mask whose values lie less than a corresponding threshold. For eg: for the *easy* level, we mask the pixels whose values are less than 0.3. By changing this threshold, we can vary the difficulty level of the masked input. We use the following thresholds: $\{easy : 0.3, medium : 0.6, hard : 0.9\}$. These thresholds were obtained by visual inspection. Intuitively, the threshold can be interpreted as the fraction of pixels that are masked at a given difficulty level – 30% being the easier case and 90% being the harder case.

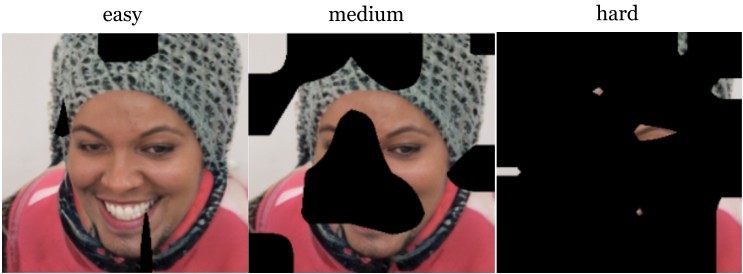

Figure 7: **Inpainting masks:** Masked inputs at different difficulty levels.

### B.3 Masking irrelevant style dimensions

In StyleGAN models, the style space vector is very high dimensional. However, previous work on style space analysis [39] has shown that only few of those dimensions are reliably disentangled. In order to better focus the encoder's capacity only on the disentangled dimensions, we mask out the irrelevant dimensions ensuring that the quantile loss is only applied to the disentangled dimensions.

For instance, for a FFHQ pretrained model trained to produce an output of size $1024 \times 1024$ has a style space dimension of 9088. In order to better focus the encoder's capacity only on the disentangled dimensions, we mask out the irrelevant dimensions. More concretely, we apply the loss function described in (9) to the masked latent, $\mathcal{L}_{q_\beta}(m \odot x, m \odot z)$, with $m$ being the mask that contains '1' for the disentangled dimensions and '0' otherwise and $\odot$ indicating element-wise product. Note that the masking is applied only to the quantile loss and not the pointwise loss in (6). This ensures that the pointwise prediction is able to match the true latents accurately, while the quantile heads focus on learning variablity only in the disentangled dimensions.

## C   Quantitative analysis of interval variability

In this experiment, we set out to measure the variability of the predicted quantile intervals as a function of problem difficulty. For this analysis, we use 500 images at each difficulty level sampled from the FFHQ-trained pretrained GAN as inputs to our encoder.

For each input, we compute the calibrated uncertainty interval using our approach and compute the *identity* loss specified in Equation 7, and *perceptual* loss between the upper and lower edges. We repeat the procedure for each image by varying the input difficulty, similar to the previous Appendix. It can be observed from Table 1 that both perceptual and ID losses increase with increasing perceived input difficulty. This substantiates our claim that the calibrated quantiles display more variability as we increase the difficulty of the task. Note that most of the style dimensions only affect attributes like hair color/glasses/facial hair that do not necessarily change the identity of the individual. Given this observation, the change in ID loss is very much indicative of the variability of the quantile predictions.

Table 1: **Measuring variability over quantiles:** Perceptual loss (L-PIPS) and ID Loss between the upper and lower calibrated quantiles.

| METRIC | EASY | MEDIUM | HARD |
|---|---|---|---|
| ID LOSS | 0.03 | 0.06 | 0.08 |
| PERCEPTUAL LOSS | 0.17 | 0.21 | 0.24 |

## D   Effect of calibration on coverage

The guarantee in Definition 2.1 tells us that the risk will always be controlled, but it does not tell us that our control will be tight. This experiment tells us how conservative our procedure is, *i.e.*, how closely we match our desired risk and error levels.

Since we work in realm of generated data for model training and calibration, we have access to the true latents $Z_d$ which ensures a precise measurement of the average risk. We do a random 50-50 split

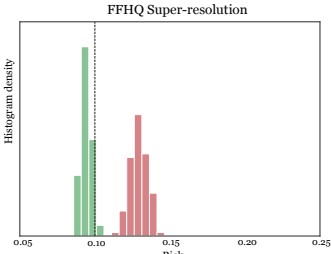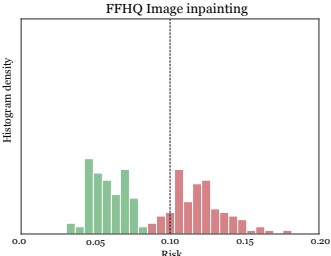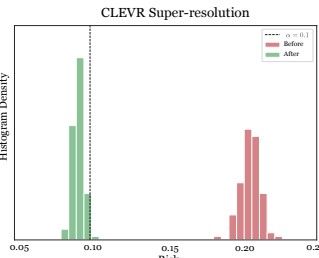

Figure 8: **Calibration:** Comparison of distribution of empirical risk for 100 calibration runs before and after performing the RCPS calibration procedure. We show results on FFHQ and CLEVR for the Image super-resolution and inpainting corruption models, calibrating for risk level $\alpha = 0.1$.

on the calibration set, where we calibrate on one split and evaluate on the other. To validate the power of the procedure, we repeat this process 100 times. For each run, we report the average risk incurred by our model over the evaluation split.

Figure 8 compares the average risk of the quantile encoders across different corruption models and datasets, before and after calibration. The performance of the uncalibrated quantile encoder is problem / dataset dependent, *i.e.*, the base model has lower risk in the FFHQ super-resolution problem compared to the inpainting problem or the CLEVR super resolution problem . However, for all settings, the calibration procedure results in lower risk that satisfies the guarantee specified in Definition 2.1.

## E  Quantifying coverage and adaptivity in real data

In Table 2, we quantify the effect of coverage and adaptivity on real data in comparison with generated data. The average set size shows that our intervals adapt to problem difficulty successfully for both real and generated data. However the value is higher in case of real data, which is expected since our encoder was not trained for quantile regression on real data. Regarding coverage metrics, we compute the average risk separtely for real and generated data, with and without calibration. While the calibration procedure does not make a significant change in the generated data, since the base encoder is has pretty good coverage to begin with, it makes a significant different in the case of real data.

Table 2: **Adaptivity and Coverage:** We measure adaptivity by computing the average set size across problem difficulty. Note that the average set size increases with problem difficulty illustrating the adaptivity of the predicted quantile intervals. For quantifying coverage, we measure the average risk before and after calibration. Note that, the risk before calibration on real data is much higher before calibration. This points to the importance of our calibration procedure especially in the presence of real data, which the encoder model was not trained on.

|  |  | **Generated data** | **Real data** |
|---|---|---|---|
| **Average set size** | Easy | 2.5 | 3.1 |
|  | Hard | 4.7 | 5.3 |
| **Average risk** | w/o calib | 0.114 | 0.267 |
|  | w/ calib | **0.085** | **0.096** |

## F  Ablation of feature loss weights

We ablate the feature loss weights that are specified in Eq 8. For simplicity, we fix $c_1 = c_2 = c$. Table 3 shows the predicted set sizes for different values of $c$. Higher values of $c$ yield more visually pleasing reconstructions during training while also providing slightly tighter quantile sets across

varying levels of corruption. Hence, we pick a value of $c = 10.0$ for our experiments. A larger hyper-parameter sweep might yield better results.

Table 3: **Ablation**: Predicted set sizes for different values of the feature coefficient, $c$. We pick the value $c = 10$ as it provides more visually pleasing reconstructions on real data.

| Corruption level | c = 0 | c = 1.0 | c = 10.0 |
|:---:|:---:|:---:|:---:|
| Easy | 2.56 | 2.4 | 2.5 |
| Hard | 5.23 | 5.1 | 4.7 |