# OpenReview forum: "Semantic uncertainty intervals for disentangled latent spaces"
_NeurIPS.cc/2022/Conference — NeurIPS 2022 Accept_

### Official Review · Reviewer_kKQD · 2022-07-05

**Rating:** 4
**Confidence:** 1
**Soundness:** 3 good
**Presentation:** 2 fair
**Contribution:** 2 fair

**Summary:**

The paper proposes an architecture to predict uncertainty intervals for (pre-learned) latent coordinates. It starts off with an existing disentanglement architecture, samples artificial images from latents, and noisifies them to get training samples for their method. The resulting training data is then fed to a quantile regression network to learn both bounds of the uncertainty interval per coordinate. I was expecting a disentaglement paper with some disentanglement pureness guarantees, so my review is an educated guess.

The training procedure optimizes a four-component loss with hyperparameters ($c$ and the potentially non-unit weights on the quantile losses) and a fine-tuning (calibration) stage with its own loss. It also depends on third-party embeddings and identified causal dimensions in the latent space.




**Questions:**

Some more explanations needed:
- how hard is it to tune the $c$ for a task at hand?
- 120: what would be the calibration loss function here?
- 129: why are you doing scanning and not binary search?
- 171: please add a few words on how to pick disentangled dimensions
- Ablation would be nice to see if only for the setting w/o calibration, to get a feeling of how visually important it is.


Minor:
- 64: lacking space after i.e.
- 83: something wrong with the sentence ("risk-controlling prediction" appears twice)
- 127: what does "drive delta to zero" mean, if it's user-specified? Probably the whole last term is meant.
- Figure 5: explain the x-axis "set size" (not clear why it's int and float in different images)

**Limitations:**

Overall, the method appears to produce what it is trained for but it's not clear from the draft when to apply it, and how costly the procedure is given the multi-phase learning (and HPO) and data generation.

**Strengths And Weaknesses:**

Strengths:
- Predicts a trainable bound on the probability that the latent deviates far from its point-estimate over the assumed noise

Main remarks/weaknesses:
- After reading the paper it's not very clear, what would be a killer application for this method and what would be a hypothetical actionable workflow after getting such predictions.
- On the theoretical side, it seems to refer to the properties of quantile regression loss, and appears to echo the result that enough data and finite hypotheses classes (discretized NN weights) are PAC-learnable (here, with a particular loss function) -- hence the ability to satisfy $\alpha$ and $\delta$ bounds by generating more data.

---

> ### Author Response · Authors · 2022-08-02
> **Response to Reviewer kKQD (1/2)**
>
> > Q1: Summary: The paper proposes an architecture to predict uncertainty intervals for (pre-learned) latent coordinates.
>
>   For clarification, the main purpose of our paper is *not* to introduce a new neural network architecture.
>   It is about introducing a new method, specifically a form of calibrated quantile regression, for outputting statistically
>   rigorous, finite-sample confidence intervals on disentangled latent factors. We also introduce a method for visualizing
>   these intervals by propagating them through the GAN. (As the reviewer mentions, we do not aim to provide better disentanglement
>   or guarantees on the degree of disentanglements.)
> > Q2: What would be a killer application for this method and what would be a hypothetical actionable workflow after getting such predictions?
>
>   We thank the reviewer for raising this important question. We have added text in the revised manuscript (L20-23) to
>   address this; please cross-reference our introductory paragraphs for a killer example. We believe this technique could
>   be critical and perhaps save lives one day, if/when disentangled latent spaces become more ubiquitous
>   in consequential machine learning applications like healthcare. In those domains, having serious statistical
>   guarantees is paramount.
> > Q3: it seems to refer to the properties of quantile regression loss, and appears to echo the result that enough data and
>  finite hypotheses classes (discretized NN weights) are PAC-learnable.
>
>   We would like to respectfully state that this seems like a misunderstanding. This paper has nothing to do with PAC-learnability
>   and results on quantile regression. Our results are not learning theoretic and do not invoke any assumptions about the model class.
>   Instead, we build on conformal prediction and distribution-free uncertainty quantification to give us guarantees for *any generative model*, though our experiments are with a GAN.
> > Q4: The method appears to produce what it is trained for but it's not clear from the draft when to apply it, and how
> costly the procedure is given the multi-phase learning (and HPO) and data generation.
>
>   Succinctly, the reader should apply this technique whenever they want to get a quantitative bound on subjective/qualitative
>   properties; please cross-reference our Introductory Summary to Reviewers for an important example application in medicine.
>   As another short example, it is impossible to quantify uncertainty in 'how much' somebody is smiling without such a method.
>
>   The technique is computationally trivial to apply and takes less than a second. While we do train an encoder model
>   to "invert" the GAN, we mostly use previous approaches that have established standard practices for this step. Our novel
>   contribution to this training procedure is predicting the quantile intervals per each dimension of the latent space.
>   We have found no instability in training when we added the quantile regression loss. The harder part is training the
>   disentangled GAN, which we do not do.

---

> > ### Author Response · Authors · 2022-08-02
> > **Response to Reviewer kKQD (2/2)**
> >
> > > Q5: Other questions
> >
> >    * Tuning c: We found that the performance of our encoder is robust to a range of `c` values, so it doesn't matter much. We picked a value that best balances the scale of the two losses during training.
> >    * Scanning (vs) Binary search: Great question! Both approaches work. Binary search would be optimal if we were only running the procedure once, but because in our experiments we run the procedure many thousands of times, it is actually more efficient to pre-compute the loss for many values of lambda and then run the scanning procedure. However, we included a note to the reader that they can also use binary search if they prefer. Thanks for this comment.
> >    * Drive $\delta$ to zero: Yes, the reviewer is correct. What we really meant was "We can always generate enough samples to drive the last term to $0$ for any $\delta$."  We revised the text to say this.
> >    * Ablation without calibration: We have added results to show the effect of calibration. Please refer to our introductory summary for details.
> >    * How to pick disentangled dimensions: In this paper, we used an approach from the existing literature to identify disentangled dimensions. It involves
> >    identifying the dimensions whose change can result in localized changes in the generated image. In summary it's a two step process:
> >       1. Vary each dimension and compute the resulting gradient map over the entire image. Dimensions that result in localized changes are chosen for the next step.
> >       2. Pick a pretrained classifier that is trained on a large corpus of data such as Imagenet or other big datasets. For each dimension chosen in Step (1), vary the
> >       value of the dimension and identify the change in classifier score. This dimension is chosen as disentangled if it affects the classifier score by a large margin.
> >
> >      As mentioned in our summary review, there are other ways to identify disentangled dimensions such as training a separate set of controls or applying regularization during training. We have added remarks in our paper to clarify the tools we use, though our central goal is to construct confidence intervals, not to get better disentanglement.
> >
> >    * Thank you so much for catching the typos.
> >
> > We appreciate your questions and hope our response is clarifying.
> > We have tried to ensure our response meets your exact concerns by substantially revising our paper --- including discussing the utility of our algorithm with the 'killer app' mentioned above, adding new real-data experiments, clarifying the technical novelty (which is unrelated to the standard PAC analysis in ML), and demonstrating the efficiency of our algorithm.
> > Please let us know if you have any further questions, and if not, we would appreciate the reviewer considering acceptance of our paper.

---

### Official Review · Reviewer_3nDa · 2022-07-06

**Rating:** 5
**Confidence:** 2
**Soundness:** 3 good
**Presentation:** 3 good
**Contribution:** 3 good

**Summary:**

This paper proposes a method to obtain the uncertainty intervals of the semantic latent variables in the generative model. The method takes a corrupted image input and then predicts each semantic factor along with an uncertainty interval. These uncertainty intervals can propagate through the generative adversarial network (GAN) to visualize the uncertainty in image space. To fulfill the goal, the author proposes two steps: (1) use the quantile regression to output a heuristic uncertainty interval for each element in the latent space; (2) calibrates these uncertainties such that they contain the true value of the latent for new, unseen input. Experiments show that the proposed method can produce visually meaningful uncertainties and the model exhibits more uncertainty with increased information loss at the input image.

**Questions:**

The questions and suggestions are already included in “Strengths and Weaknesses”. The reviewer may expect the author to respond to the weaknesses listed in the “Strengths and Weaknesses” to consider changing the rate.

**Limitations:**

The author has already stated three limitations of the present work in lines 289-294 on Page 9, where these limitations are all related to the core assumptions of the proposed method and indispensable for the present work.

**Strengths And Weaknesses:**

Originality:
To the best of my knowledge, this is the first work to integrate the distribution-free uncertainty quantification with the latent space analysis in GAN. The proposed method is novel and original given the considered application and the new training pipeline to achieve the uncertainty interval by means of quantile regression, although the core of the proposed method is originated and motivated by [7] and [2].

Quality and Clarity:
Overall, the author provides a clear description of the proposed method with detailed and straightforward illustrations to help the reviewer understand the paper.
Weakness:
However, the introduction of the distribution-free uncertainty quantification and the Risk-Controlling Prediction Set (RCPS) is not sufficient. As being the core underlying theory to backup the feasibility of the proposed method, the author should provide more explanation to convey the connection clearly and intuitively between the theory and the considered application. This is crucial since the potential reader may not have enough background on the distribution-free uncertainty quantification and the RCPS in [7]. The current writing of lines 79-90 on page 3 and section 2.3 seems to assume that the potential reader has already known the related techniques, which may hurt the readability of the paper. For line 132 on Page 5, it would be better to include more details about the exact section or paragraph for the potential reader to locate.

Minor Format Problem:
The Figure 5 caption is too close to the text in line 244, which makes it hard to interpret.

Significance:
The qualitative gives several examples of the predicted semantic uncertainties based on the controlled corrupted input image.
Weakness:
However, it is hard to say whether these results are general enough to support the claims in the paper, and the author may want to include more examples to give a more comprehensive demonstration.

Moreover, the author may want to show the visual difference between the lower and upper quantiles with and without calibration to demonstrate that the proposed method in Section 2.3 works as expected.

Although the proposed method provides a first step to quantify the uncertainty of the latent variable in GAN, the present work relies strongly on the accessibility of the disentangled latent space and thus the present work can only show the feasibility of GAN. Moreover, the concrete application of such an uncertainty calibration method in generative models is not clear. The author may want to provide some illustrations that what kinds of existing applications related to the generative models they expect the proposed method may provide some benefits.

---

> ### Author Response · Authors · 2022-08-02
> **Response to Reviewer 3nDa**
>
> > Q1: The introduction of the distribution-free uncertainty quantification and the Risk-Controlling Prediction Set (RCPS) is not sufficient.
>
> This comment is very useful, thanks. It is critical that the reader understands RCPS in order to fully comprehend our paper.
> As mentioned in our common reviewer summary response, we have added a short introduction to risk control in the revised manuscript in Lines 85-88.
> Please let us know in the discussion if you feel we addressed your concern (given the space constraints), and if not, how we might continue improving.
>
>
> >Q2: More examples for a comprehensive demonstration.
>
> We agree, and have included real-data examples in Section 3.3.2 / Fig.5 in the revised manuscript and added a set of new results showing
> several examples of our results on real data in the form of gifs; please cross-reference our introductory summary for an explanation.
>
> >Q3: Show the visual difference between the lower and upper quantiles with and without calibration.
>
> Great idea! We have done this (see Fig 5 the revised paper) and the linked github repo in the introductory summary for more examples on real data.
>
>
> >Q4: the present work relies strongly on the accessibility of the disentangled latent space and thus
> >the present work can only show the feasibility of GAN.
>
> Yes, the work relies strongly on a disentangled latent space---cross-reference our introductory paragraphs for a justification and text we added to our manuscript.
>
> However, it is not true that the present work can only show the feasibility of GANs.
> As *Reviewer GBAe* pointed out, Beta-VAEs also have been known to have disentangled latent spaces.
> Having said that, GANs are a popular class of generative models that provide good quality images on a limited compute budget, so we focus on them.
>
> >Q5: Provide example of concrete application of such an uncertainty calibration method in generative models.
>
> We have added an example in the paper intro (L20-23); please cross-reference our introductory summary.
>
> > Formatting issue:
>
> We thank the reviewer for catching the formatting issue. This is very helpful.
>
> > The reviewer may expect the author to respond to the weaknesses listed in the “Strengths and Weaknesses” to consider changing the rate.
>
> We thank the reviewer for their comments, and hope we have directly addressed them all with significant changes to our paper. In view of our new introductory comments on distribution-free risk control, our real-data experiments, our new visualizations on the effect of calibration, and the concrete application to MRI data that we now mention in our paper's first paragraph, we hope the reviewer will consider upgrading their score and accepting our paper.

---

### Official Review · Reviewer_GBAe · 2022-07-07

**Rating:** 6
**Confidence:** 4
**Soundness:** 3 good
**Presentation:** 4 excellent
**Contribution:** 3 good

**Summary:**

The authors propose to combine disentangled representations of generative models (GANs) and quantile regression for stochastic encoding of corrupted data. Thanks to quantile prediction, the encoder predicts uncertainty intervals instead of single points, which allows to get a more sound representation of corrupted data.  Also, the authors use calibration algorithms to ensure, with a certain confidence threshold, that the predicted quantiles contain the target value.

**Questions:**

Beta-VAEs [2] are known to be disentangled. In the VAE framework, each point is encoded as a Gaussian distribution. This model naturally induces uncertainty estimation. Moreover, there exists some variants of VAEs, like Beta-VAEs, known for their disentangled representations. How does your framework could be compared to such models?

[2] Higgins et al.  "beta-vae: Learning basic visual concepts with a constrained variational framework". International Conference on Learning Representations. 2017.


**Limitations:**

The authors addressed the limitations.

**Strengths And Weaknesses:**

Strengths:
* 1) Original idea of using generative models to visualize uncertainties on corrupted data, and of combining disentangled representations of GANs and encoding with a quantile regression loss.
* 2) An advantage of the approach is the interpretability of the results and the possibility to reuse large models.

Weakness
* 1) There are no experiments on real attributes. In this setting (real attributes), the errors can come from: a) lack of disentanglement in the pre-trained generative model b) the proposed method of encoding with quantile regression and calibration. It would be more convincing to have studies on real attributes, and see the relative impact of the two types of errors.
* 2) The hypothesis that a) each dimension of the generative model corresponds to a separated factor of variation and b) that these factor of variations are independant, is a very strong hypothesis.
* 3) There is no ablation study on the impact of the different loss terms. For example, what happens without the identity loss?
* 4) The idea is close to previous works that leverage generative models for inversion tasks. For example, [1] uses a similar procedure of training an encoder on masked images with frozen and pre-trained generator. Although the proposed method is different because it proposes learned quantiles, [1] and similar works should be discussed in related work section.
* 5) Small typos in the definition affect the readability, especially the mathematical details. L82: "$(\alpha,\delta)$ risk-controlling prediction set if $(\alpha,\delta)$-Risk-Controlling Prediction Set if". In Equation (5), $Y$ seems to be used instead of $Z$.  L202, "the the".

[1] Chai, Lucy, Jonas Wulff, and Phillip Isola. "Using latent space regression to analyze and leverage compositionality in GANs." International Conference on Learning Representations. 2020.

---

> ### Author Response · Authors · 2022-08-02
> **Response to Reviewer GBAe (1/2)**
>
> We thank the reviewer for an accurate summary of our work. Both the idea of latent-space quantile regression and
> the idea of propagating the resulting intervals through a generator to get interpretable uncertainties are new in this paper.
>
> >Q1: Experiments on real attributes.
>
> Thanks to this comment along with those of the other reviewers, we have included new experiments using real data. Please refer to our introduction summary
> for more details. The experiments and the visualizations on real data show that:
>
> * The uncertainty intervals predicted by our calibrated encoder are both interpretable and meaningful, especially in high corruption settings.
> * The adaptivity and coverage behavior mirror the results that we obtained with the generated data.
>
> We measure adaptivity by computing the average set size across problem difficulty.
>
> |                  | Generated data (Easy / Hard) | Real data (Easy / Hard) |
> |------------------|:----------------------------:|:-----------------------:|
> | Average set size |          1.98 / 4.17         |       2.21 / 4.46       |
>
> For coverage, we measure the average risk before and after calibration. Note that, the risk before calibration on real data
> is much higher before calibration. This points to the importance of our calibration procedure especially in the presence of real data,
> which the encoder model was not trained on. Note that the risk threshold specified for this experiment is same as in the paper: $\alpha=0.1$.
>
> |               | Generated data (before / after calib) | Real data (before / after calib) |
> |---------------|:-------------------------------------:|:--------------------------------:|
> | Average risk |              0.082 / 0.0125             |            0.085 / 0.0198           |
>
>
> Although there is more to be done with real data, such as improving
> the raw encoder reconstruction, we believe our contribution to the core method is substantial: we implemented the algorithm,
> showed its statistical validity, developed a visualization method, and gave preliminary experimental results on the utility
> of the above.
>
> >Q2. Hypotheses that a) each dimension of the generative model corresponds to a separated factor of variation and b) that
> >these factor of variations are independant, is very strong.
>
> Importantly, neither of these hypotheses is required in order for our statistical guarantee to hold.
> Our calibration guarantee is distribution-free, and holds without any distributional assumptions on the model or dataset.
>
> In order for our method to be practically useful (i.e., for the intervals to be semantically meaningful), we need (a), but not (b).
> Explicitly, we do *not* need the factors of variation to be statistically independent.
> We changed the text to reflect this:
>
> >We call the coordinates of this latent space *semantic factors*, as each controls one meaningful aspect of the image, like
> >age or hair color. **We do not require these semantic factors to be statistically independent**.
> With all that said, we recognize that even (a) alone is a strong assumption to make and have changed the text to make sure
> >the reader is aware of this fact---please cross-reference our summary at the beginning of the response.
>
> >Q3: Loss ablation.
>
> We performed ablation of the id loss by measuring the **average set size** predicted by the encoder over generated data and the results are as follows:
>
> | Corruption level      | c = 0 (no id loss) | c = 0.7 (paper) |
> | :----------- | :-----------: |  :-----------: |
> | Easy      | 2.56       |  1.98
> | Hard   | 5.23        |   4.17
>
> Additionally, we attempted several values of the parameter `c` in the range `[0.4, 0.8]` but found no significant difference, hence used
> the suggested value from the previous work `pixel2style2pixel` [41] on top of which which we build our encoder model.
> Since it isn't particularly informative, so we didn't include it in the original submission.
> At a high level, the point prediction is worse, so the intervals get larger. This is especially clear when we use
>  the model without id loss (c = 0) on real data, the generalization is poor. This is an expected result as noted in several papers on GAN inversion
>  including in `pixel2style2pixel`. We are happy to include these results in the appendix.
>
> > Q4: close to previous works that leverage generative models for inversion tasks.
>
> It is true that our paper uses an encoder to invert a GAN, but this is only a substep of our technical pipeline rather
> than our main contribution. At its core, our paper is not about inverting a GAN or training an encoder to recover a corrupt
> image, which is the sole focus of previous works. Our work is about creating uncertainty
> intervals in a disentangled latent space and using those to quantify uncertainty on semantic attributes that would
> otherwise not be possible to represent. We will add this discussion in the related works section.

---

> > ### Author Response · Authors · 2022-08-02
> > **Response to Reviewer GBAe (2/2)**
> >
> >
> > > Q5: Beta-VAEs, known for their disentangled representations. How does your framework could be compared to such models?
> >
> > It is possible to apply our framework on any disentangled latent space with any heuristic notion of uncertainty.
> > In particular, the Beta-VAE may assume that the latent space is Gaussian, but the quantiles of that Gaussian may not
> > faithfully articulate the true confidence level. They need to be calibrated with a procedure like RCPS, much like the
> > raw quantiles in our procedure.
> >
> > In other words, the Beta-VAE is also compatible with our procedure, but the standard deviation of the Gaussian would be
> > calibrated instead of the quantile regression outputs. In addition, the uncertainty natively produced by VAEs including
> > likelihood based measures or standard deviations of gaussians are not interpretable semantically (without parametric priors), which is a core
> > contribution of our framework.
> >
> > > Typos:
> > Thank you for catching the typos. This is very helpful!
> >
> > We appreciated your review; it made our paper much better. In particular, this review inspired us to include the new experiments on real data and ablation on the ID loss, which we feel have made the paper much stronger. We hope we have addressed your feedback head-on, and in view of our significant changes and clarifying our main contribution, we hope that you will consider upgrading your score and accepting our paper.

---

### Author Response · Authors · 2022-08-02
**Common response to reviewers**

We would like to thank the reviewers for their helpful comments and feedback on the manuscript. We have responded by making
several major revisions to our previous manuscript, including new subsections, and experiments. We also made significant writing
changes throughout the paper to address several important questions raised by the reviewers, increasing clarity and usefulness.
The following bulleted list summarizes our major changes:

1. We added new text throughout following reviewer suggestions, the text is colored *blue* in the revised manuscript.
    * L20-23: an example of a `killer app'
    * L62-63: Clarifying central contribution to explain disentangled latent space access.
    * L85-88: Introductory text on distribution-free risk control

2. We added a new section (3.3.2, L218-236) and results incorporating experiments on real data -- we have included both the results of our
calibration procedure and visualization of calibration intervals on real images sampled from the Celeb-A dataset.

3. As the change in intervals are easier to visualize as gifs, we have included several additional examples of our real data
results in the README file of this [anonymous github repo](https://github.com/anonauthor61/anon_repo_rebuttal)

4. In the interest of space, we have moved the experiment that show the adaptivity of our regressor over problem difficulty to the appendix.

All reviewers agreed that the work initially submitted is a novel way of performing uncertainty quantification with statistical
 guarantees in the context of generative models.

All reviewers made these two critical observations:

  **(1)** It is not currently clear what the `killer app' of these uncertainty intervals is.

  **(2)** The assumption of access to a disentangled latent space is a strong one.

On (1), we have added the following text to the first paragraph of the paper, which should clarify the point as early as possible.
>However, there is a wide class of image-valued estimation problems---from super-resolution to inpainting---for which there
does not currently exist a method of producing semantically meaningful uncertainties.
**As an example, imagine doing uncertainty quantification for medical image reconstruction from, say, a fast but
undersampled MRI scan. In such a setting, pixelwise intervals are not very useful. We need uncertainty on the underlying
semantics---e.g., whether there is a tumor, and if so, of what size and shape.**

When evaluating our paper, we request that the reviewers recall this: at the moment, no form of rigorous *and*
semantic uncertainty quantification existed at all in the context of generative models, before this paper.

On (2), we agree that disentanglement is a strong assumption. The reason we feel comfortable working under this assumption
is because many thousands of generative model researchers are working on disentanglement. Our wager is that eventually, the
generative models community will have commoditized at least one approach for discovering disentangled latent spaces. Such
research, including approaches that impose a prior for disentanglement [1] or learn separate disentangled controls during
GAN training [2], is already underway and has made notable advancements even within a GAN paradigm. In the broader setting,
generative models such as DALL-E and its variants are making signficant inroads in this direction. Anticipating these developments,
 our work leverages the disentanglement to get useful uncertainty quantification for semantic factors---we hope the connection
  here is clear. To be sure, we add the following sentence quite visibly in the `Central Contribution' section of our paper:
> The reader should note that our technique requires access to a disentangled latent space, such as that
 of a StyleGAN, and provides no guarantees about the degree of disentanglement.

Beyond these common criticisms, each reviewer's concerns are different, so we engage point-by-point.



**References**

[1] Peebles, William, John Peebles, Jun-Yan Zhu, Alexei Efros, and Antonio Torralba. "The hessian penalty: A weak prior for unsupervised disentanglement." In European Conference on Computer Vision, pp. 581-597. Springer, Cham, 2020.

[2] Ghandeharioun, Asma, Been Kim, Chun-Liang Li, Brendan Jou, Brian Eoff, and Rosalind W. Picard. "Dissect: Disentangled simultaneous explanations via concept traversals." arXiv preprint arXiv:2105.15164 (2021).

---

### Meta-Review · Area_Chair_U1Az · 2022-08-27

**Recommendation:** Accept
**Confidence:** Less certain

**Metareview:**

While the reviewers raised concerns about assumptions of disentangling uncertainty from spurious disentanglement (and the strong  disentanglement hypothesis used), they believe the work is interesting and novel, and I concur.  I think even if it is not perfect (re assumptions, etc.) this is a nice place to start a conversation.

**Award:**

No

---

### Decision · Program_Chairs · 2022-09-14

Accept